# Mitochondrial proteins: Potential pathophysiological mechanisms of malignant progression in HCC

Yicun Liu[1☯], Yawen Shao[2☯], Xudong Zhu[3], Zhouming Shen[1], Zhaolian Bian[1]*

**1** Department of Gastroenterology and Hepatology, Nantong Third People's Hospital, Affiliated Nantong Hospital 3 of Nantong University, Nantong, Jiangsu, China, **2** Nantong University Medical School, Nantong, Jiangsu, China, **3** Department of Gastroenterology, Nantong First People's Hospital, Affiliated Nantong Hospital 2 of Nantong University, Nantong, Jiangsu, China

☯ These authors contributed equally to this work.
* bianzhaolian1998@163.com

## Abstract

### Objective

Based on the special role of mitochondria in tumour energy metabolism. We hope to explore the pathogenesis and potential therapeutic targets of Hepatocellular carcinoma by analysing the expression of 1136 mitochondrial proteins in hepatocellular carcinoma and their mechanisms in the Human.MitoCarta3.0 database.

### Methods

The expression of 1136 mitochondrial proteins in HCC was analysed by the TCGA database. We selected the top eight mitochondrial proteins among the highly expressed mitochondrial proteins that had not been studied in HCC and were statistically ($P < 0.05$) significant, according to fold change. Protein expression was verified by real-time quantitative reverse transcription polymerase chain reaction in tumours and adjacent paracancerous tissues of 34 pairs of HCC patients. Further in HCC cells, the expression of FDPS, DNA2 and MYO19 was verified. Clinical correlations of FDPS, DNA2 and MYO19 were analysed by UALCAN and KM-plot databases. Immune correlation of FDPS, DNA2 and MYO19 was analysed by TIMER2.0 and Sangerbox3.0 online databases.

### Results

Mitochondrial proteins were expressed on all 24 chromosomes. More than 2/3 of the mitochondria were 100–600 bp long, of which 204 were secondary transmembrane proteins. 1136 mitochondrial proteins, of which 202 are not included in the TCGA database. Of the 934 mitochondrial proteins included in the TCGA database, 706 were highly expressed and 228 were poorly expressed in HCC. Further validated by

**Data availability statement:** All the data used in this article are from Human MitoCarta3.0: 1136 mitochondrial Genes database (https://personal.broadinstitute.org/scalvo/MitoCarta3.0/human.mitocarta3.0.html) and UCSC Xena database (https://xena.ucsc.edu). The data in both databases can be downloaded for free.

**Funding:** This study was supported by grants from Nantong Science and Technology Bureau (JC2023115), Six peak talents in Jiangsu Province (YY-177), Youth Fund of Natural Science Foundation of Jiangsu Province (BK20200965), Scientific Research Fund of Nantong Health Commission (MB2020037), Nantong Science and Technology Bureau (MSZ2024117), and Health Bureau of Nantong City (MB2021057). The funders had no role in study design, data collection and analysis, decision to publish, or preparation of the manuscript.

**Competing interests:** All authors have declared that no competing interests exist.

HCC tissues and cells, the study found that significantly high expression of FDPS, DNA2 and MYO19 was negatively correlated with the prognosis of HCC patients. The results of the immune correlation analysis showed that DNA2 and MYO19 may be involved in regulating the infiltration of immune cells.

## Conclusion

934 out of 1136 mitochondrial proteins in the Human.MitoCarta3.0 database were differentially expressed in HCC, suggesting that mitochondrial proteins play an important biological role in the development of HCC. Further experimental validation and bioinformatics analyses showed that functional mitochondrial proteins are potential pathophysiological mechanisms for malignant progression of HCC. Mitochondrial proteins, in the future, have the potential to be valuable therapeutic targets for HCC.

## 1. Introduction

Hepatocellular carcinoma (HCC) is the predominant type of primary liver cancer, accounting for approximately 75–85% of cases. HCC is one of the major cancers of all malignant tumours in which mortality far outweighs morbidity [1,2]. Due to its malignant proliferative nature, cancer has always been considered a proliferative disease. More recently, there is growing evidence that it is a metabolic disease [3,4]. Imbalances in cellular energy metabolism are as prevalent in cancer cells as other features of cancer function (growth inhibition escape, sustained proliferation, inhibition of apoptosis, unrestricted cell replication, induction of neovascularisation and invasion and migration) [5,6].

Metabolic reprogramming is widely recognized as a hallmark of malignancies and serves as an important adaptation mechanism for HCC cells to microenvironment [6,7]. Targeting energy metabolism has become a crucial direction in cancer treatment [8]. As a key organelle, mitochondria play an essential role in regulating tumor cell metabolism [9]. Mitochondria are functional organelles with a double membrane structure, which play a fundamental role in basic biological functions such as material metabolism, energy generation, ion storage and cell proliferation regulation [10]. First, liver cells are rich in mitochondria, which make up 13–20 percent of the liver's volume [11]. Secondly, a large number of studies have found that extensive mitochondrial abnormalities are observed in the histology of liver fibrosis, cirrhosis, and primary liver cancer [10,12,13].

Mitochondrial proteins participate in controlling the proliferation, apoptosis, and metabolic reprogramming of tumor cells, making them better adapted to the tumor microenvironment [14]. Mitochondrial dynamin proteins, such as mitofusin 1 (MFN1) and mitofusin 2 (MFN2), play crucial roles in different types of tumors. Research by Huang et al. found that MFN1 is positively correlated with the overall survival (OS) of HCC patients [15]. Rehman et al. found that downregulation of MFN2 promotes mitochondrial fission, thereby enhancing tumor (lung cancer) proliferation, reducing apoptosis, and facilitating malignant progression [16]. Studies have shown that the

mitochondrial-associated protein DNA damage-inducible transcript 4 (DDIT4) is elevated under stress conditions such as chemotherapy, hypoxia, and DNA damage. Abnormally increased DDIT4 is involved in various malignant biological behaviors, including tumor drug resistance, malignant proliferation, and invasion [17].

This study focuses on HCC and mitochondrial proteins to excavate the biological functions of mitochondria in the development of HCC. First, initial screening and validation was performed through relevant common databases on mitochondria, tumor and immunity. Then, through further experimental validation and mechanism mining, it provides the basis for further basic and clinical experiments. We hope that our study will provide a new direction for the study of HCC-associated mitochondrial proteins and improve the status quo of diagnosis and treatment of HCC, in which the mortality rate of HCC is higher than the morbidity rate.

## 2. Materials and methods

### 2.1. Workflow

All the workflows are shown in Fig 1.

### 2.2. Mitochondrial protein data download and correlation analysis

Human MitoCarta3.0: 1136 mitochondrial Genes database (https://www.broadinstitute.org/files/shared/metabolism/mitocarta/human.mitocarta3.0. html) was used to download data related to 1136 mitochondrial proteins. The expression of 1136 mitochondrial proteins in HCC was analysed using the TCGA visualisation database UALCAN (http://ualcan.path.uab.edu/). We first obtained the gene sequences of the mitochondrial proteins through NCBI (https://www.ncbi.nlm.nih.gov). The mitochondrial proteins were then characterised for their transmembrane properties through the TMHMM 2.0 website (https://services.healthtech.dtu.dk/services/TMHMM-2.0/).

### 2.3. Gene Ontology (GO) and Kyoto Encyclopedia of Genes and Genomes (KEGG) enrichment analyses

We performed GO and KEGG enrichment analysis on 706 high expressed mitochondrial proteins. GO and KEGG enrichment analysis and image visualisation were done through the online database Bioinformatics (http://bioinformatics.com.cn/basic_tumor_purity_estimation_by_estimate_t016).

### 2.4. Collection of tissue specimens

Thirty-four patients with HCC (≥18 years old) who underwent radical surgical treatment at Nantong Third People's Hospital affiliated with Nantong University were included in the study (From February 1, 2021 to February 1, 2022). The patients' HCC tumours and their paired paracancerous tissues were collected. Specimens were transferred via liquid nitrogen after ex vivo and stored in a −80 °C degree refrigerator. The study was approved by the Ethics Committee of the Third People's Hospital of Nantong, affiliated with Nantong University, and each subject signed an informed consent form (Ethics Approval No. EL2021012).

### 2.5. Cell lines and culture conditions

The human normal cell line LO2 and the human HCC cell line Hep3B2.1-7 were purchased from the Stem Cell Bank of the Chinese Academy of Sciences (Shanghai, China), and were uniformly cultured in a humidified incubator containing 5% $CO_2$ at a temperature of 37°C ($CO_2$–311, Thermo).

### 2.6. Quantitative reverse transcription polymerase chain reaction (qRT-PCR) of tissues and cells

HTseq-Counts gene expression data of 374 HCC patients and 50 normal subjects in the TCGA-LIHC25 cohort were downloaded from the UCSC Xena database (https://xena.ucsc.edu). The top 20 genes upregulated in HCC were selected

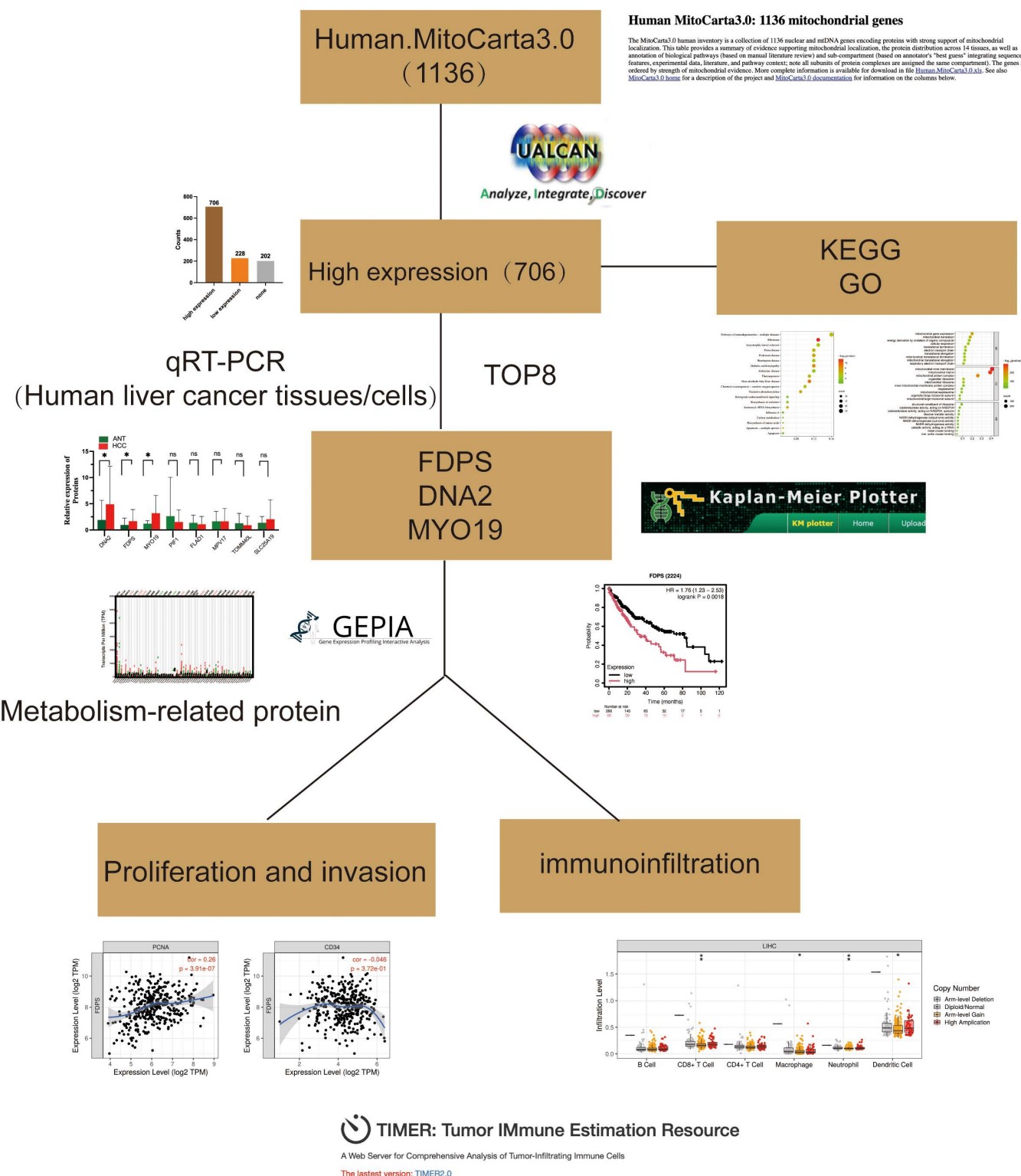

**Fig 1. Flow chart of the study.** GO: Gene Ontology; KEGG: Kyoto Encyclopedia of Genes and Genomes; qRT–PCR: quantitative reverse transcription polymerase chain reaction.

according to the differential expression ratio to make heat maps. The verified genes were excluded, and the first 8 mitochondrial proteins that were highly expressed in HCC were selected for qRT-PCR validation.

RNAiso Plus (Takara, Beijing, China) was used to extract tissue and cellular total RNA. total RNA was reverse transcribed to cDNA according to the instructions of PrimeScript RT Master Mix (Takara). gene amplification and detection were performed using a TB Green Premix Ex Taq II (Takara) and a CFX Connect fluorescence quantitative PCR instrument (BIO-RAD, USA). Relative expression was analyzed using the comparative cycle threshold (Ct) ($2^{-\Delta\Delta Ct}$) method. All experiments were performed in triplicate. Table 1 lists the glyceraldehyde 3-phosphate dehydrogenase (GAPDH) and eight mitochondrial protein primer sequences.

### 2.7. Clinical correlation and immunological correlation analysis of FDPS, DNA2 and MYO19

The expression levels of FDPS, DNA2 and MYO19 in HCC were analysed by UALCAN and the human protein Atlas (https://www.proteinatlas.org). UALCAN and KM plot (http://kmplot.com/analysis/) were used to analyse the prognostic and clinical relevance of mitochondrial proteins. FDPS, DNA2 and MYO19 immune cell correlation and infiltration analyses were performed using TIMER 2.0 and Sangerbox 3.0 (http://www.sangerbox.com/home.html).

### 2.8. Statistical analysis

GraphPad Prism 8.0 and SPSS 24.0 are used for data analysis and image visualization. All data were presented as the mean ± standard deviation (SD). Statistical significance was defined as $P < 0.05$.

## 3. Results

### 3.1. Mitochondrial protein characteristic analysis in HCC

Most of the 1136 mitochondrial proteins were 100–600 bp long (Fig 2A), and more than half were expressed on both the inner and outer membrane of the mitochondria (Fig 2B), evenly distributed on 24 human chromosomes (Fig 2C).

**Table 1. Primer sequences of TOP 8 Mitochondrial protein.**

| Primer | Primer sequence (5'--3') |
| --- | --- |
| GAPDH-F | CAATGACCCCTTCATTGACC |
| GAPDH-R | TTGATTTTGGAGGGATCTCG |
| PIF1-F | CACGGTTCTGCTTCCAGTCCAAG |
| PIF1-R | TCTGCCTGCCTCCACACCTTG |
| TOMM40L -F | TGTATCCAGGGTGTTCCAGGTGAG |
| TOMM40L-R | CTGCCATGATGCCGCTGCTC |
| DNA2-F | CGCCTGGTCATCACTGCTTCAC |
| DNA2-R | CCTGGCTCTACTGGAACAGAACAC |
| FDPS-F | CGGGAGAGGTGGCTGGGTTC |
| FDPS-R | TCATTCTGAGGGAGGAGCAAAGGG |
| FLAD1-F | CCACCCTCCCTGCTCCAGATAC |
| FLAD1-R | TCCGCTCCTCTTCTTCGTTCTCC |
| MPV17-F | CTGGAAGGCACATCGGCTCTAAG |
| MPV17-R | GCCCACTTTGAGGAGGTTGTCTG |
| SLC25A19-F | CAGTGAGAAGGAGCAGGAAGTTG |
| SLC25A19-R | TGATAGGTGTAGGATGGCGTCTG |
| MYO19-F | AGCCGCAGTCCAGACCTACC |
| MYO19-R | CTCGTCCTCACTGGCTCCTTTG |

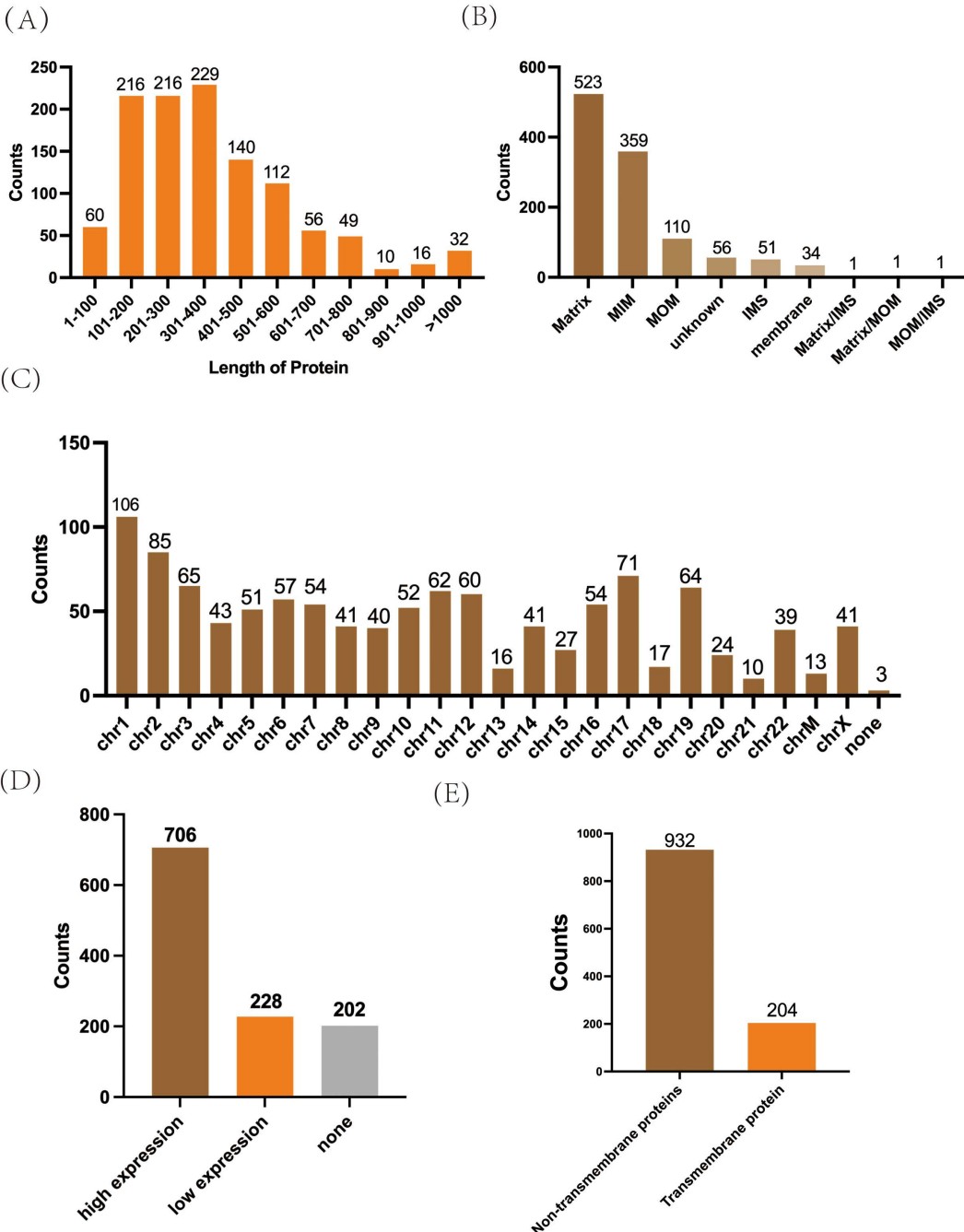

**Fig 2. Basic characteristics and expression of mitochondrial proteins in HCC. (A)** Amino acid number of mitochondrial proteins; **(B)** Cellular location of mitochondrial proteins; **(C)**. Distribution of mitochondrial proteins in human chromosomes; **(D)**. mitochondrial proteins differentially expressed in HCC (202 are not included in TCGA); **(E)**. Number of transmembrane proteins in mitochondrial proteins.

Verification results at TCGA showed that 706 out of 1136 mitochondrial proteins were highly expressed in hepatocellular carcinoma (Fig 2D). An online database analysis showed that 204 of 1136 mitochondria were transmembrane proteins (Fig 2E).

### 3.2. KEGG and GO enrichment analysis of mitochondrial protein in HCC

GO enrichment analysis of the high expressed mitochondrial proteins included three categories: cellular component, biological process, and molecular function. In the cellular component analysis, the three most enriched terms were mitochondrial inner membrane, mitochondrial matrix and mitochondrial protein complex. In the biological process analysis, mitochondrial gene expression, mitochondrial translation, energy derivation by oxidation of organic compounds were the three most enriched terms. In the molecular function analysis, structural constituent of ribosome, oxidoreductase activity, acting on NAD(P)H and oxidoreductase activity, acting on NAD(P)H, quinone were the most prominent terms (Fig 3A). KEGG pathway enrichment analysis showed that the functions of the high expressed mitochondrial proteins may be focused on Oxidative phosphorylation, Non-alcoholic fatty liver disease and Diabetic cardiomyopathy (Fig 3B).

### 3.3. Expression and prognosis of mitochondrial protein in HCC

Highly expressed mitochondrial proteins in HCC were selected based on TCGA raw data according to fold change (FC). The top 20 mitochondrial proteins with FC > 2.5 (MY019, SLC25A19, ACLY, MPV17, RAB24, POLQ, PIF1, DNA2, MAVS, DTYMK, RECQL4, MTHFD1L, NT5DC2, HPDL, PYCR1, OX6B2, TOMMAOL, FLAD1, FDPS and COX412) were used as study objects to produce heat maps (Fig 4A). We selected the first 8 highly expressed mitochondrial proteins that had not been validated in HCC (MY019, SLC25A19, MPV17, PIF1, DNA2, TOMMAOL, FLAD1 and FDPS) and validated them in 34 pairs of HCC tissues, which showed that MY019(FC ≈ 2.7), DNA2(FC ≈ 2.5), and FDPS(FC ≈ 1.7) were significantly overexpressed in HCC tissues (Fig 4B, P < 0.05). We further verified the expression levels of MY019, DNA2 and FDPS in HCC cells. The results showed that MY019(FC ≈ 2.0), DNA2(FC ≈ 2.3) and FDPS(FC ≈ 1.9) were significantly highly expressed in the human hepatocellular carcinoma cell line Hep3B2.1-7 compared to LO2 in normal human hepatocytes (Fig 4C, P < 0.05).

### 3.4. Pan-cancer analysis of FDPS, DNA2 and MYO19

Pan-cancer analysis showed that FDPS was significantly highly expressed in HCC, bladder cancer, breast cancer, colon cancer, esophageal cancer, head and neck squamous cell carcinoma, renal clear cell carcinoma, lung adenocarcinoma,

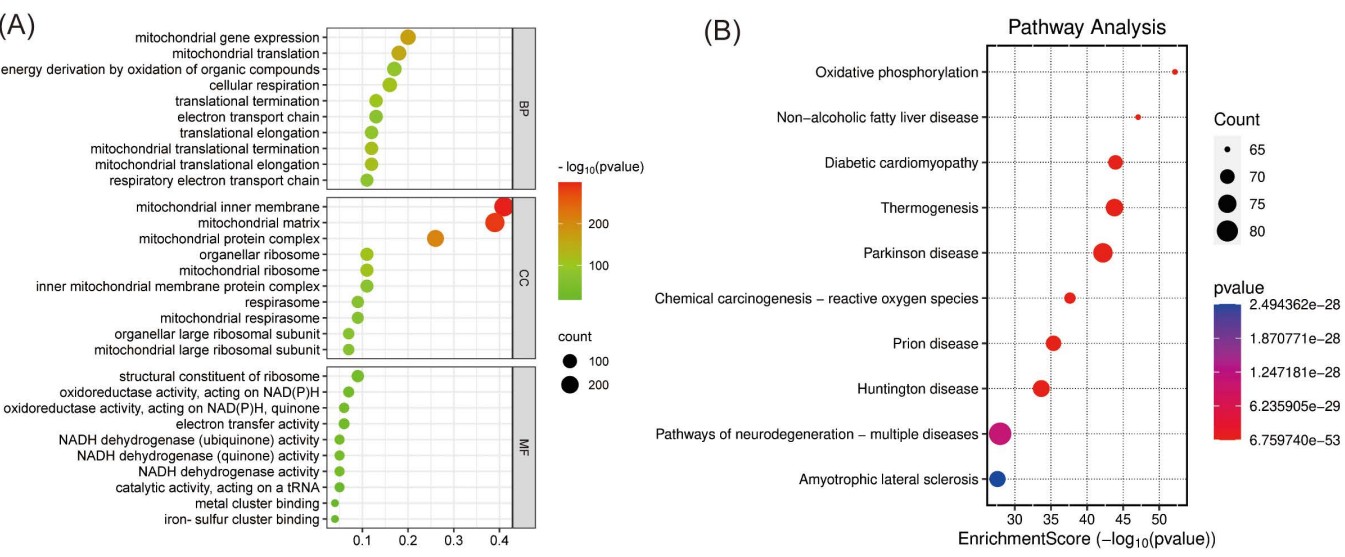

**Fig 3. Gene Ontology (GO) and Kyoto Encyclopedia of Genes and Genomes (KEGG) enrichment analyses. (A)**. GO enrichment analysis of 706 mitochondrial proteins highly expressed in HCC; **(B)**. KEGG enrichment analysis of 706 mitochondrial proteins highly expressed in HCC.

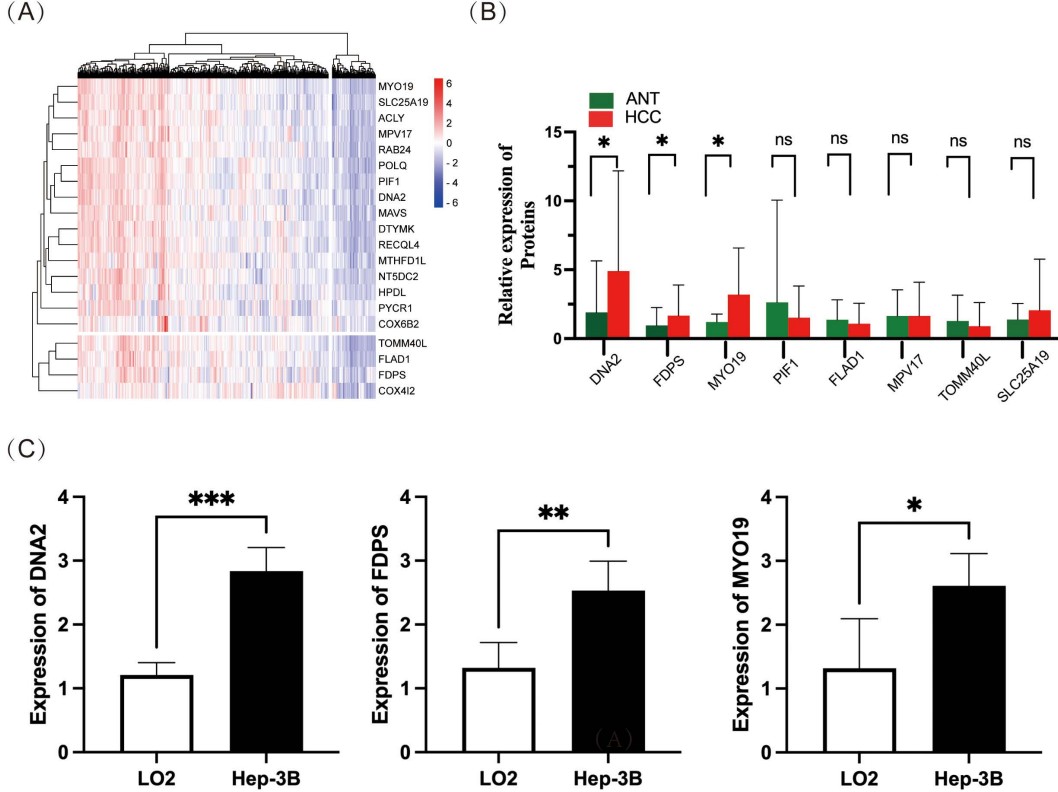

**Fig 4. Screening of differentially expressed mitochondrial proteins. (A)**. Heatmap showing the top 20 (order of fold change) mitochondrial proteins highly expressed in HCC; **(B)**. qRT-PCR validation of the top 8 mitochondrial proteins highly expressed in HCC (n = 34); **(C)**. qRT-PCR validation of FDPS, DNA2 and MYO19 expression in HCC cells.

non-small cell lung carcinoma, gastric adenocarcinoma and endometrial cancer. In addition, FDPS was lowly expressed in papillary cell carcinoma of the kidney, while there was no difference in the expression in thyroid and prostate adenocarcinomas (Fig 5A, P > 0.05). In addition to melanoma and thyroid cancer, DNA2 is highly expressed in these other tumors (Fig 5B). MYO19 is underexpressed in thyroid and renal clear cell carcinoma, and highly expressed in the remaining tumors (Fig 5C).

We further analyzed the multiplicity of change of FDPS, DNA2 and MYO19 in HCC by the UALCAN database, which were about 3, 3.5 and 3.8, respectively (Fig 5D, 5G and 5J). Explore the cellular distribution, major functions and protein interaction networks of the three mitochondrial proteins via The human protein Atlas website.FDPS is primarily associated with cellular metabolism and interacts with six proteins (SLC30A2, SSMEM1, RNF19B, ABHD16A, EIF-4ENIF1 and ATXN1) (Fig 5E). FDPS was predominantly distributed in the cytoplasm, with a small amount expressed in the nucleus (Fig 5F). DNA2 is a key enzyme involved in DNA replication and DNA repair in the nucleus and mitochondria, and may serve as a target for drug action, and is closely related to 4 proteins (PTGES3, CIAO1, MMS19 and CIAO2B) (Fig 5H). DNA2 is mainly distributed in mitochondria (Fig 5I). MYO19 is localized in the outer mitochondrial membrane and may be involved in mitochondrial transport and localization, interacting with three proteins (RHOT2, AKAP1 and FBXL4, Fig 5K). In addition, MYO19 was predominantly distributed in mitochondria, with a small amount expressed in the cytoplasm (Fig 5L).

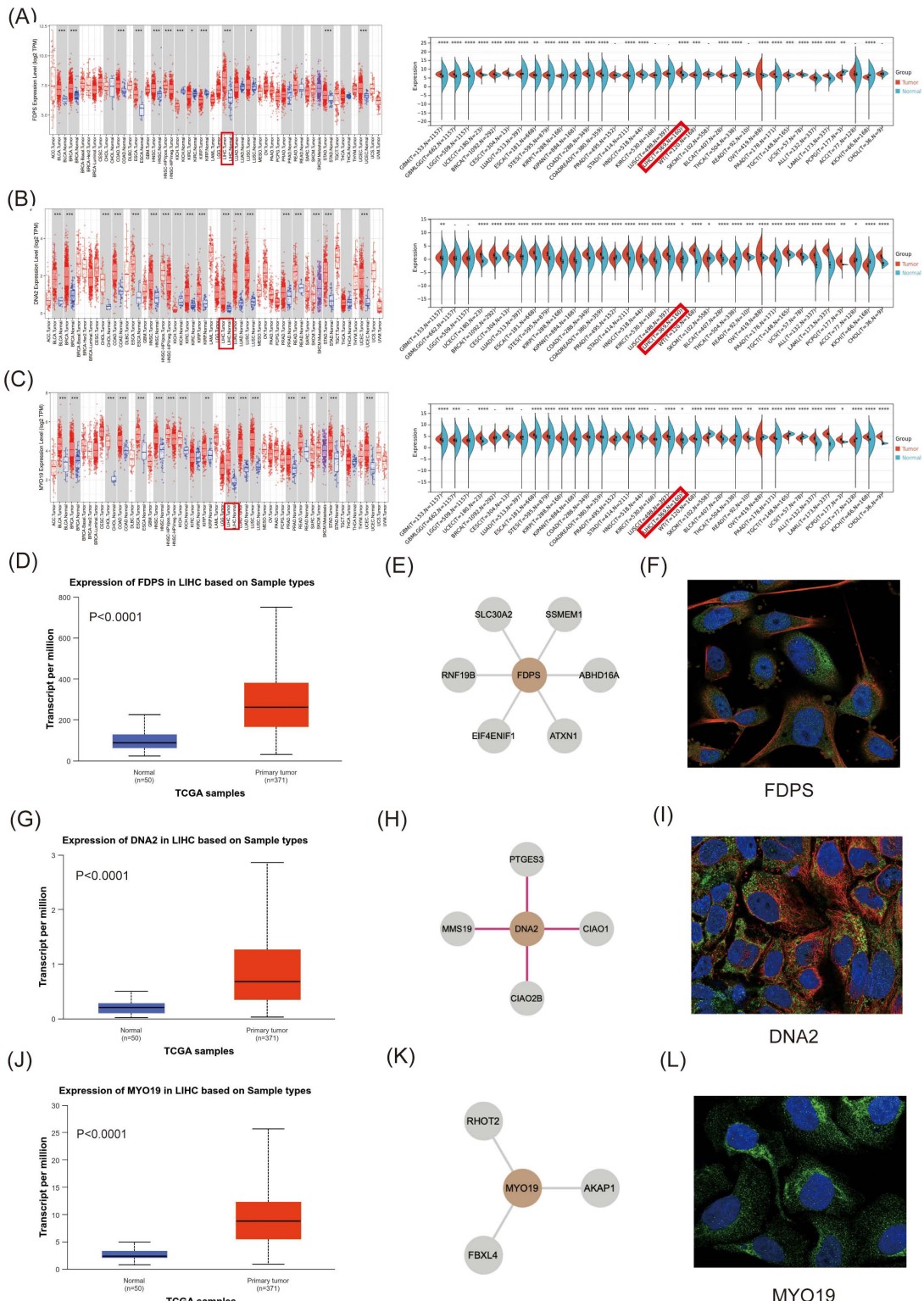

**Fig 5. Pan-cancer analysis of FDPS, DNA2 and MYO19. (A).** Pan-cancer analysis of FDPS; **(B).** Pan-cancer analysis of DNA2; **(C).** Pan-cancer analysis of MYO19; **(D).** FDPS was highly expressed in HCC; **(E).** FDPS protein interaction network diagram; **(F).** Distribution of FDPS in cells; **(G).** DNA2 was highly expressed in HCC; **(H).** DNA2 protein interaction network diagram; **(I).** Distribution of DNA2 in cells; **(J).** MYO19 was highly expressed in HCC; **(K).** MYO19 protein interaction network diagram; **(L).** Distribution of MYO19 in cells.

### 3.5. Clinical relevance of FDPS, DNA2, and MYO19 in patients with HCC

We used the UALCAN database to investigate the relationship between FDPS, DNA2 and MYO19 and clinical parameters in HCC patients. Patients with HCC were divided into subgroups based on age, gender, lymph node metastasis status, TP53 mutation status, tumor grade, and individual cancer stage. We found significantly elevated levels of FDPS, DNA2, and MYO19 expression at all ages (Fig 6A). There were no significant differences in FDPS expression levels with respect to patient gender, lymph node metastasis status, and TP53 mutation status. The expression level of DNA2 was significantly upregulated in women, lymph node metastasis and TP53 mutation groups of HCC patients. There were no differences in gender, lymph node metastasis, and MYO19. The expression of MYO19 was significantly upregulated in HCC patients in the TP53 mutant group (Fig 6B, 6C and 6D). According to tumor grade, the expression levels of three mitochondrial proteins were significantly higher in grades 1, 2, 3 and 4 compared to controls. The expression levels of DNA2

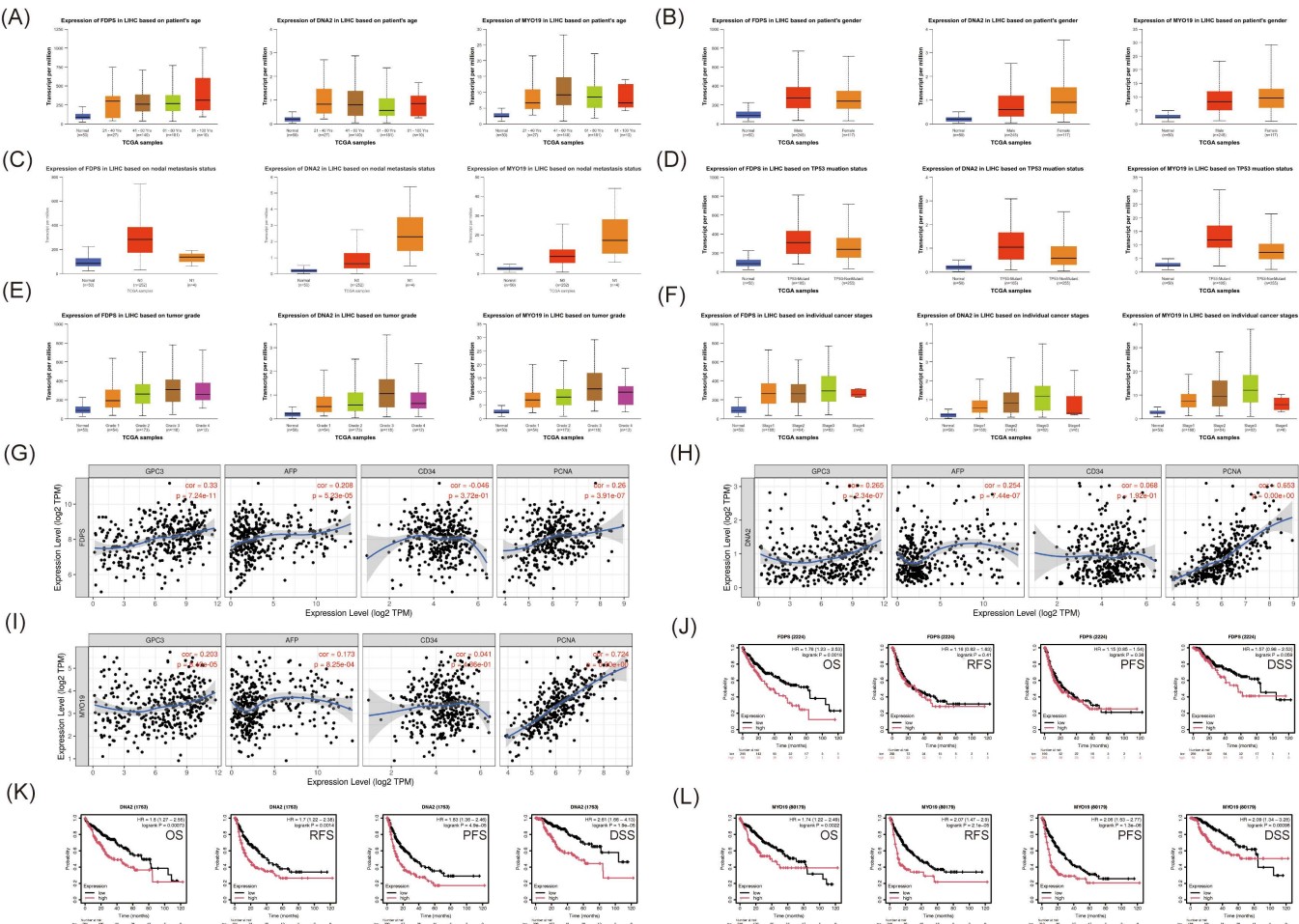

**Fig 6. Clinical correlation analysis of FDPS, DNA2 and MYO19. (A)**. Differential expression of FDPS, DNA2 and MYO19 in HCC patients by age; **(B)**. Differential expression of FDPS, DNA2 and MYO19 in HCC patients by gender. **(C)**. Differential expression of FDPS, DNA2 and MYO19 in lymph node metastasis of HCC patients; **(D)**.Differential expression of FDPS, DNA2 and MYO19 in TP53 mutations of HCC patients; **(E)**.Differential expression of FDPS, DNA2 and MYO19 in various tumour grades of HCC patients; **(F)**.Differential expression of FDPS, DNA2 and MYO19 expression differences in various tumour stages in HCC patients; **(G)**.Correlation analysis of FDPS with AFP, GPC3, PCNA and CD34; **(H)**. Correlation analysis of DNA2 with AFP, GPC3, PCNA and CD34; **(I)**. Correlation analysis of MYO19 with AFP, GPC3, PCNA and CD34; **(J)**. Survival analysis analysis of FDPS with prognosis of HCC patients; **(K)**. Survival analysis of DNA2 with prognosis of HCC patients; **(L)**. Survival analysis of MYO19 with prognosis of HCC patients.

and MYO19 were relatively higher in HCC patients with tumor grade 3 compared with tumor grades 1 and 2 (P < 0.05, Fig 6E). The expression levels of DNA2 and MYO19 were significantly upregulated in FDPS at stages 1, 2, 3, and 4 according to individual cancer stage. Among them, the expression levels of DNA2 and MYO19 were upregulated in stages 2 and 3 compared with stage 1 (P < 0.05, Fig 6F).

Correlation analysis showed that all three mitochondrial proteins were significantly positively correlated with AFP, GPC3 and PCNA. However, there was no correlation between any of the three proteins and CD34 (Fig 6G, 6H and 6I). We used the KM-Plotter database to analyze the effect of three mitochondrial proteins on the prognosis of HCC patients. HCC patients were categorized into high and low expression groups based on median protein expression. Survival analysis showed that patients with lower expression of FDPS, DNA2 and MYO19 had better overall survival (OS, Fig 6J, 6K and 6L). Among them, patients with lower DNA2 and MYO19 expression also had better disease-specific survival (DSS), progression-free survival (PFS) and relapse-free survival (RFS) (Fig 6K and 6L).

These results suggest that the mitochondrial proteins FDPS, DNA2 and MYO19 are closely associated with clinical parameters and prognosis of HCC patients.

### 3.6. Immune correlation of FDPS, DNA2 and MYO19 in HCC

We analyzed the immune-related data of FDPS, DNA2 and MYO19 using the TIMER database. The results showed no significant correlation between FDPS and various immune cell populations (Fig 7A). There was a significant correlation between the expression of DNA2 and MYO19 and the infiltration of various immune cell populations, including B cells, T cells (CD8+ and CD4+), neutrophils, macrophages, and dendritic cells (Fig 7B and 7C). In addition, we observed that copy number mutations in DNA2 partially inhibited neutrophil infiltration. Copy mutations in MYO19 inhibited infiltration of CD8+ T cells, macrophages, neutrophils and dendritic cells. The results suggest that DNA2 and MYO19 may be involved in regulating immune cell infiltration (Fig 7D and 7E). To further assess the impact of DNA2 and MYO19 on patient prognosis and tumor mutational load, we evaluated stromal, immune and ESTIMAT scores (Fig 7F and 7G). Our findings provide evidence that DNA2 and MYO19 are negatively correlated with the above three scores. This suggests that increased expression of DNA2 and MYO19 accelerates the development of HCC.

## 4. Discussion

1136 mitochondrial proteins, of which 706 showed high expression in HCC, which may be associated with the high proliferative properties of the tumor. Statistical analysis in this study revealed that 80% of mitochondrial proteins are 100–600 amino acids long. The common belief is that the longer the protein, the higher the chance of mutation. Proteins between 50 and 500 amino acids long are more stable [18]. We found that FDPS with a length of 419 amino acids was more stable by comparing protein expression in both TP53 mutant and non-mutant groups of patients. DNA2 and MYO19 were 1060 and 970 amino acids long, respectively, with significant differences in expression between the two groups of patients. FDPS is a key enzyme in cholesterol biosynthesis and has been found to play an important role in the development of several malignant tumors [19,20]. Our study found that mitochondrial protein FDPS is highly expressed in HCC and negatively correlated with prognosis (OS). Numerous previous studies have shown that FDPS is strongly associated with tumor drug resistance (For example, FDPS modulates platinum sensitivity in human ovarian cancer [21]; Disruption of the FDPS/Rac1 axis increases the sensitivity of radiation therapy for pancreatic ductal adenocarcinoma [19]). Therefore, we hypothesized that FDPS may exert a biological function by modulating chemotherapeutic drug sensitivity in HCC patients.

Mitochondria and immune cells interact closely [22]. Mitochondria are functional organelles with a double membrane structure, which play a fundamental role in basic biological functions such as material metabolism, energy generation, ion storage and cell proliferation regulation [10]. Recent studies have shown that mitochondria are not only a source of energy but also play a crucial role in regulating cell death signaling and innate immunity.[23]. Our findings indicate that the expression of mitochondrial MYO19 and DNA2 is significantly associated with the tumor infiltration of various immune

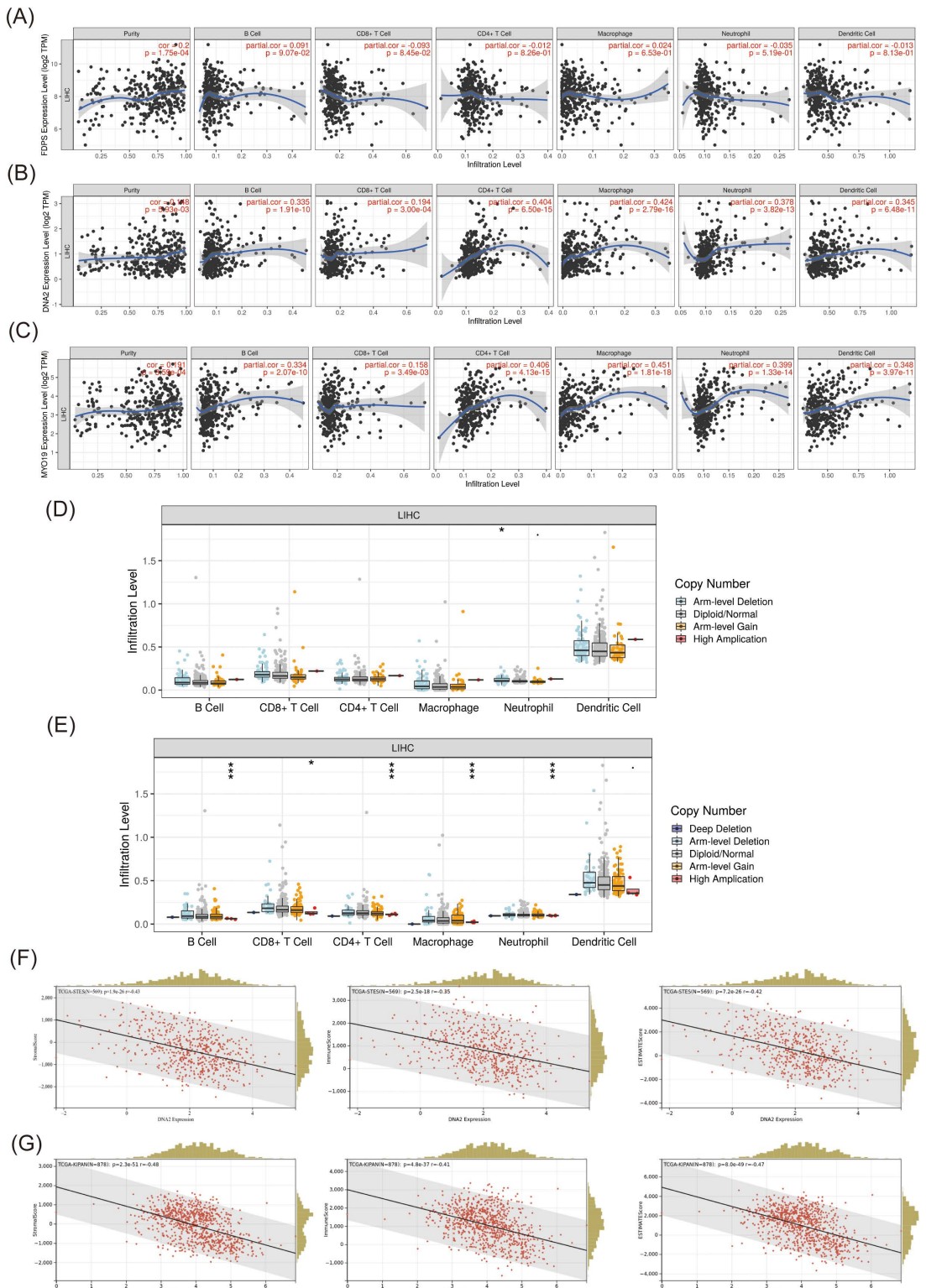

**Fig 7. Immunocorrelation analysis of FDPS, DNA2 and MYO19. (A)** Correlation analysis between FDPS and each immune cell; **(B)** Correlation analysis between DNA2 and each immune cell; **(C)** Correlation analysis between MYO19 and each immune cell; **(D)** Effect of DNA2 copy number mutant fraction on each immune cell; **(E)** Effect of MYO19 copy number mutant fraction on each immune cell; **(F)** Matrix, immune and ESTIMAT scores of DNA2; **(G)** Matrix, immunity and ESTIMAT scores of MYO19.

cells, including B cells, T cells (CD8+ and CD4+), neutrophils, macrophages, and dendritic cells. Based on the results of immune correlation analyses, we hypothesize that mitochondrial proteins may regulate tumor progression by influencing the immune microenvironment. In summary, mitochondrial proteins play crucial biological roles in the development and progression of HCC, warranting further basic and clinical investigations.

The mitochondrial repair protein DNA replication deconjugase/nuclease 2 (DNA2) mediates genome stability. It plays an important role in maintaining the functional integrity of mitochondria. Our study showed that DNA2 showed high expression in most tumors, including HCC, except for melanoma and thyroid tumors. Clinical correlation results showed that DNA2 was negatively correlated with OS, DSS, PFS and RFS in HCC. With tumor development, the expression of DNA2 gradually increased. In summary, DNA2 plays an important role in the malignant progression of HCC. In addition, DNA2 expression was associated with multiple immune cell infiltrations, indicating a very strong tumor relevance. Furthermore, copy number mutations in DNA2 partially inhibited neutrophil infiltration and did not affect other immune cell infiltration. Studies have shown that neutrophils can infiltrate tumors and participate in malignant progression of tumors through their phenotypic and functional plasticity [24,25]. Tumor-associated neutrophils (TAN) with a pro-carcinogenic phenotype are involved in all stages of tumorigenesis and development, including tumor initiation, metastasis, and immunosuppression [26]. Increased neutrophil infiltration in solid tumors is usually associated with a poor prognosis in patients with tumors [27]. Therefore, we hypothesized that DNA2 may accelerate the malignant biological progression of HCC by accelerating TAN infiltration through genetic mutations.

Myosin (MYO) is the only known motor of actin. Many studies have highlighted their dysregulation in cancer as an important factor influencing tumor invasion and apoptosis [28]. Our results showed that MYO19 was significantly highly expressed in HCC tissues and cells. In addition, MYO19 showed a significant positive correlation with antiproliferative cell nuclear antigen (PCNA) (correlation coefficient: 0.724) and a negative correlation with prognosis. Next, we will focus on the effects of MYO19 on the malignant functions of HCC proliferation, invasion and apoptosis. As with DNA2, MYO19 expression is associated with infiltration of a variety of immune cells. Differently, copy number mutations in MYO19 not only partially inhibited neutrophil infiltration, but also increased tumor infiltration by CD8+ T cells, macrophages and dendritic cells. Our findings indicate that MYO19 promotes the malignant biological progression of HCC and may be closely related to the tumor microenvironment (TME). Studies have shown that mitochondria play a crucial role in regulating the immunity of infiltrating immune cells in TME [29].Infiltrating immune cells (T cells, natural killer cells, and macrophages) are metabolically reprogrammed by mitochondria to be more adaptive to the unfavorable conditions of TME and to enhance their antitumor activity [30]. Recent studies have shown that targeting mitochondria can inhibit tumor progression by improving immune cell function [31,32]. Taken together, targeting the immune strong infiltration-associated mitochondrial protein MYO19 may provide a new direction for immunotherapy of HCC.

Our study explored for the first time the expression levels of all mitochondrial proteins in HCC. Most mitochondria are less susceptible to mutation, have higher stability, and are expected to be promising therapeutic targets and biomarkers for HCC. We also validated three mitochondrial proteins, MYO19, DNA2 and FDPS, which are closely associated with HCC development through cell and tissue experiments. In addition, mitochondria are likely to accelerate the malignant progression of HCC by affecting the tumor microenvironment/metabolic reprogramming, etc. Mitochondrial proteins play an important biological importance in HCC disease progression and deserve further investigation.

## 5. Conclusions

In summary, our study identified a large number of mitochondrial proteins that are highly expressed in HCC. In addition, we identified three (MYO19, DNA2 and FDPS) mitochondrial proteins that are closely associated with poor prognosis of HCC. The specific pathologic mechanisms by which mitochondrial proteins influence the development of HCC await more detailed studies in the future.

## Supporting information

**S1 Data. Raw data.**

(XLSX)

## Acknowledgments

We thank the Institute of Liver Diseases of Nantong Third People's Hospital for providing the experimental platform.

## Author contributions

**Data curation:** Yawen Shao, Zhouming Shen.

**Formal analysis:** Xudong Zhu.

**Funding acquisition:** Zhaolian Bian.

**Investigation:** Yawen Shao.

**Methodology:** Yicun Liu.

**Project administration:** Zhaolian Bian.

**Writing – original draft:** Yicun Liu.

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
