## [Decision Letter · Decision Letter 0]

4 Jun 2025

PONE-D-25-20385Mitochondrial proteins: potential pathophysiological mechanisms of malignant progression in HCCPLOS ONE

Dear Dr. Bian,

Thank you for submitting your manuscript to PLOS ONE. After careful consideration, we feel that it has merit but does not fully meet PLOS ONE’s publication criteria as it currently stands. Therefore, we invite you to submit a revised version of the manuscript that addresses the points raised during the review process.

We look forward to receiving your revised manuscript.

Kind regards,

Xinjun Lu

Academic Editor

PLOS ONE

Additional Editor Comments (if provided):

Reviewers' comments:

Reviewer's Responses to Questions

**Comments to the Author**

1. Is the manuscript technically sound, and do the data support the conclusions?

Reviewer #1: Yes

Reviewer #2: Yes

2. Has the statistical analysis been performed appropriately and rigorously? 

Reviewer #1: Yes

Reviewer #2: I Don't Know

3. Have the authors made all data underlying the findings in their manuscript fully available?

Reviewer #1: Yes

Reviewer #2: Yes

4. Is the manuscript presented in an intelligible fashion and written in standard English?

Reviewer #1: Yes

Reviewer #2: Yes

5. Review Comments to the Author

Reviewer #1: Modest sample size but well-conducted study that highlights the prognostic significance of MYO19, DNA2, and FDPS in HCC using human samplese and cellular lines

No bibliographic evidence available on the subject

Reviewer #2: Liu et al have documented the expression of mitochondrial proteins in HCC, out of which three proteins were significantly associated with prognosis. The manuscript is well written. There is no mention about the histological subtypes (Macrotrabecular massive, Steatohepatitic, Fibrolamellar, etc) of HCC included in the study. Was the histology reviewed 34 patients?

Some minor changes: Figure 6. J,K,L are survival analysis instead of correlation analysis.

6. PLOS authors have the option to publish the peer review history of their article (what does this mean? ). If published, this will include your full peer review and any attached files.

**Do you want your identity to be public for this peer review?** For information about this choice, including consent withdrawal, please see our Privacy Policy .

Reviewer #1: No

Reviewer #2: No

---

## [Author Response · Author response to Decision Letter 1]

16 Jun 2025

Dear Editors and Reviewers:

On behalf of my co-authors, we thank you very much for giving us an opportunity to revise our manuscript. We appreciate you very much for your valuable comments on our manuscript entitled “Mitochondrial proteins: potential pathophysiological mechanisms of malignant progression in HCC”. Your comments are very helpful in improving our paper, and they are significant guidance to our researchers. We have studied the suggestions carefully and have provided point-by-point responses in the letter below. We hope our responses adequately all editorial concerns and those of the reviewers, and that our manuscript is now acceptable for publication in your journal.

Review Comments to the Author

Reviewer #1: Modest sample size but well-conducted study that highlights the prognostic significance of MYO19, DNA2, and FDPS in HCC using human samplese and cellular lines No bibliographic evidence available on the subject.

Response: We are very grateful for your professional advice.Our understanding of the roles of MYO19, DNA2 and FDPS in the prognosis of HCC was mainly obtained through databases such as KM-plot. We are very sorry. Due to the relatively small sample size we verified (34 pairs), we are temporarily unable to use our samples for survival analysis. We have been continuously collecting specimens and plan to conduct further in-depth research.

Reviewer #2: Liu et al have documented the expression of mitochondrial proteins in HCC, out of which three proteins were significantly associated with prognosis. The manuscript is well written. There is no mention about the histological subtypes (Macrotrabecular massive, Steatohepatitic, Fibrolamellar, etc) of HCC included in the study. Was the histology reviewed 34 patients?

Response: First of all, thank you very much for your recognition and suggestions on this article. I consulted the histological classification of the pathology department of our hospital. Among the 34 pairs of HCC tissues, 25 cases were Macrotrabecular massive, 5 cases were adenoid, and 4 cases were Steatohepatitic. No special histological classification was found. We previously only focused on the pathological classification and never paid attention to the histological classification. We will strengthen the research on the histological classification in the future.

Some minor changes: Figure 6. J,K,L are survival analysis instead of correlation analysis.

Response: Thank you very much for the comments. We are very sorry for such a mistake in the article. We have corrected the incorrect representation in the captions (P23).

---

## [Decision Letter · Decision Letter 1]

14 Jul 2025

Mitochondrial proteins: potential pathophysiological mechanisms of malignant progression in HCC

PONE-D-25-20385R1

Dear Dr. Bian,

We’re pleased to inform you that your manuscript has been judged scientifically suitable for publication and will be formally accepted for publication once it meets all outstanding technical requirements.

Kind regards,

Xinjun Lu

Academic Editor

PLOS ONE

Reviewers' comments:

Reviewer's Responses to Questions

**Comments to the Author**

1. If the authors have adequately addressed your comments raised in a previous round of review and you feel that this manuscript is now acceptable for publication, you may indicate that here to bypass the “Comments to the Author” section, enter your conflict of interest statement in the “Confidential to Editor” section, and submit your "Accept" recommendation.

Reviewer #1: All comments have been addressed

Reviewer #2: All comments have been addressed

2. Is the manuscript technically sound, and do the data support the conclusions?

Reviewer #1: Yes

Reviewer #2: Yes

3. Has the statistical analysis been performed appropriately and rigorously? 

Reviewer #1: Yes

Reviewer #2: Yes

4. Have the authors made all data underlying the findings in their manuscript fully available?

Reviewer #1: Yes

Reviewer #2: Yes

5. Is the manuscript presented in an intelligible fashion and written in standard English?

Reviewer #1: Yes

Reviewer #2: Yes

6. Review Comments to the Author

Reviewer #1: (No Response)

Reviewer #2: The manuscript is well written and the authors have appropriately responded to the suggestions with revision.

7. PLOS authors have the option to publish the peer review history of their article (what does this mean? ). If published, this will include your full peer review and any attached files.

**Do you want your identity to be public for this peer review?** For information about this choice, including consent withdrawal, please see our Privacy Policy .

Reviewer #1: No

Reviewer #2: No

---

## [Editor Report · Acceptance letter]

PONE-D-25-20385R1

PLOS ONE

Dear Dr. Bian,

I'm pleased to inform you that your manuscript has been deemed suitable for publication in PLOS ONE. Congratulations! Your manuscript is now being handed over to our production team.

Kind regards,

on behalf of

Dr. Xinjun Lu

Academic Editor

PLOS ONE